# Imaging Evaluation of Intervertebral Disc Degeneration and Painful Discs—Advances and Challenges in Quantitative MRI

**DOI:** 10.3390/diagnostics12030707

**Published:** 2022-03-14

**Authors:** Shota Tamagawa, Daisuke Sakai, Hidetoshi Nojiri, Masato Sato, Muneaki Ishijima, Masahiko Watanabe

**Affiliations:** 1Department of Medicine for Orthopaedics and Motor Organ, Juntendo University Graduate School of Medicine, 2-1-1 Hongo, Bunkyo-Ku, Tokyo 113-8421, Japan; s-tamagawa@juntendo.ac.jp (S.T.); hnojiri@juntendo.ac.jp (H.N.); mi@juntendo.ac.jp (M.I.); 2Department of Orthopaedic Surgery, Surgical Science, Tokai University School of Medicine, 143 Shimokasuya, Isehara 259-1193, Kanagawa, Japan; sato-m@is.icc.u-tokai.ac.jp (M.S.); masahiko@is.icc.u-tokai.ac.jp (M.W.); 3Center for Musculoskeletal Innovative Research and Advancement (C-MiRA), Tokai University Graduate School, 143 Shimokasuya, Isehara 259-1193, Kanagawa, Japan

**Keywords:** intervertebral disc degeneration, discogenic low back pain, T2 mapping, diffusion-weighted MRI, ADC mapping, T1ρ mapping, sodium MRI, q-space imaging, CEST, MR spectroscopy

## Abstract

In recent years, various quantitative and functional magnetic resonance imaging (MRI) sequences have been developed and used in clinical practice for the diagnosis of patients with low back pain (LBP). Until now, T2-weighted imaging (T2WI), a visual qualitative evaluation method, has been used to diagnose intervertebral disc (IVD) degeneration. However, this method has limitations in terms of reproducibility and inter-observer agreement. Moreover, T2WI observations do not directly relate with LBP. Therefore, new sequences such as T2 mapping, T1ρ mapping, and MR spectroscopy have been developed as alternative quantitative evaluation methods. These new quantitative MRIs can evaluate the anatomical and physiological changes of IVD degeneration in more detail than conventional T2WI. However, the values obtained from these quantitative MRIs still do not directly correlate with LBP, and there is a need for more widespread use of techniques that are more specific to clinical symptoms such as pain. In this paper, we review the state-of-the-art methodologies and future challenges of quantitative MRI as an imaging diagnostic tool for IVD degeneration and painful discs.

## 1. Introduction

The number of patients with low back pain (LBP) in daily medical practice is extremely high, and it is estimated that 70–85% of people will experience LBP at some point in their lives [1]. Disability due to LBP is a significant socioeconomic problem, and the total cost of LBP in the United States is estimated to exceed 100 billion USD annually [2]. The causes of LBP are multifactorial, but often originate from the tissues that comprise the spinal column, such as intervertebral discs, facet joints, sacroiliac joints, nerve roots, and paraspinal muscles [3]. Intervertebral disc (IVD) degeneration is an independent risk factor for LBP [4], and the frequency of discogenic LBP has been reported to be around 26–42% of all LBP [3,5,6]. Furthermore, as IVD degeneration progresses, disc herniation, lumbar spinal canal stenosis, and lumbar degenerative spondylolisthesis may occur, causing not only LBP but also lower-extremity radicular pain, numbness, muscle weakness, and, in the worst case, paralysis [7,8]. Therefore, it is critical to accurately diagnose IVD degeneration and provide appropriate prevention and treatment.

While some reports [9,10] have shown that IVD degeneration as observed on imaging is not always associated with LBP, imaging has an essential role in the diagnosis and is crucial in determining changes over time, as well as response to treatment. Among imaging methods, magnetic resonance imaging (MRI) is the most frequently used modality for the evaluation of IVD degeneration in clinical practice because it can capture various anatomical and physiological changes in vivo in a minimally invasive manner. The Pfirrmann classification on T2-weighted imaging (T2WI) is the most widely used method for evaluating IVD degeneration by MRI [11]. However, this method is a qualitative evaluation method based on the examiner’s visual interpretation, and it has limitations in terms of reproducibility, inter-observer agreement, and correlation with clinical symptoms [12]. Therefore, there is a need for more widespread use of techniques that are more specific to clinical symptoms such as pain, which are more useful for diagnosis, treatment, and prognosis. In addition, more objective modalities to evaluate IVD regeneration and alleviation of pain are needed for the development and assessment of the upcoming regenerative medical products, such as cell therapy [13,14], gene therapy [15,16], and tissue engineering [17,18]. In recent years, MRI sequences that enable various quantitative evaluations have been developed with the aim of overcoming these problems. The purpose of this paper was to describe these contemporary quantitative MRI techniques, i.e., T2 mapping, diffusion-weighted imaging (DWI) with apparent diffusion coefficient (ADC) mapping, T1ρ mapping, sodium MRI, q-space imaging (QSI), chemical exchange saturation transfer (CEST), and magnetic resonance spectroscopy (MRS), and to discuss their potential and limitations toward the evaluation of IVD degeneration and painful discs.

## 2. Mechanism of IVD Degeneration

### 2.1. Structure and Characteristics of IVD

The IVDs connect the adjacent upper and lower vertebral bodies, dispersing and supporting the load placed on the spinal column; with their superior flexibility, they enable flexion, extension, and rotation. The IVD is composed of the nucleus pulposus (NP), annulus fibrosus (AF), and cartilaginous endplate (CEP), each having different structural and biochemical properties [7]. The NP in the center of the IVD occupies about half the volume of the disc and retains relatively large quantitates of water due to its extracellular matrix (ECM) structure consisting of type II collagen fibers and proteoglycans (PGs), comprising about 80% water (wet weight basis) in a healthy state [7]. The AF surrounding the NP is composed of 15–25 concentric lamellae, in which the collagen fiber orientation differs for each adjacent lamella, imparting mechanical strength to this structure [19]. The inner layer of the AF is rich in type II collagen, while the outer layer is a fibrocartilaginous tissue rich in type I collagen. The CEP is connected to the vertebral body, sandwiching the top and bottom of the IVD, and it is responsible for most of the nutrient supply to the IVD by diffusion, making the IVD the largest avascular structure in the human body [20,21].

### 2.2. Mechanism of Degeneration

IVD degeneration can occur earlier in life than most other musculoskeletal structures, beginning in the 11–16 year old age group [22]; structural lumbar disc degeneration is observed in 40% of those under 30 years of age and in 90% of those over 50 years of age [23]. Although the full degeneration cascade remains elusive and is likely heterogeneous, the most common biochemical change observed is a loss of PGs within the NP area [24,25]. As a result, the osmotic pressure of the IVD decreases and water retention is reduced. Although the absolute amount of collagen in the IVD changes little, the type and distribution of collagen changes with age and degeneration, with a decrease in the synthesis of type II collagen and an increase in the synthesis of type I collagen [26]. Matrix-degrading enzymes such as cathepsins, matrix metalloproteinases, and aggrecanases are also involved in IVD degeneration [27]. Structurally, the border between NP and AF is blurred as the water content in the NP decreases with PG loss, leading to a low signal intensity on T2WI and to a decrease in the disc height, with the disc becoming increasingly disorganized as degeneration progresses [7]. In addition, reduced diffusion due to mechanical stress, aging, and degeneration-induced CEP stiffening and calcification can easily lead to nutrient deficiencies within the disc, resulting in a low-glucose, hypoxic, and highly acidic environment that accelerates IVD degeneration by triggering cellular senescence and cell death [20,28,29].

## 3. Clinical Features and MRI Diagnosis of Discogenic LBP

Discogenic LBP patients tend to have increased pain sensation during forward bending movements and sitting positions, likely due to an increase in intradiscal pressure [30]. Its diagnosis optimally includes a comprehensive determination through imaging evaluations such as X-ray radiography and MRI, as well as direct manipulation of the recurrent pain through discography or discoblock, in addition to physical findings [31]. Sociopsychological factors may also play a role in the chronicity of pain, and it is important to evaluate these comprehensively [1,32]. Although reports have shown that IVD degeneration findings on imaging are not always associated with LBP and vice versa [9,10], MRI is a minimally invasive modality that can capture various anatomical and physiological changes in the body and is the most frequently used modality to determine the extent of IVD degeneration and the potential source of pain. Here, we introduce different MRI observations and classifications of the routinely clinically acquired T2WI and their potential impact for identifying discogenic pain.

### 3.1. T2-Weighted Imaging (T2WI)

T2WI is most common MRI sequence and is based on the proton density and relaxation time. Relaxation is the process via which a nuclear “spin” returns to thermal equilibrium after absorbing radiofrequency energy. Tissue can be characterized by two different relaxation times—T1(longitudinal relaxation) and T2 (transverse relaxation) [33]. T2WI is an imaging technique that emphasizes the differences in T2 relaxation phenomena by imaging at the echo time that provides the best tissue-to-tissue contrast. Therefore, the signal intensity varies depending on which echo time is used, and a more quantitative method is needed.

### 3.2. Pfirrmann Classification

The Pfirrmann classification, an evaluation method for lumbar disc degeneration using MRI T2-weighted midsagittal images, was reported by Pfirrmann et al. [11] in 2001, and it is at present the most widely used classification for evaluating human disc degeneration. It is a five-grade qualitative evaluation method that focuses on the relative T2 signal intensity and homogeneity of the NP, as well as the maintenance of NP-AF distinction and disc height (Figure 1) [34]. This classification was devised from a relatively young cohort and proved suboptimal to classify the progression of degeneration, particularly in elderly patients with advanced degeneration. Accordingly, Griffith et al. [12] proposed a modified Pfirrmann classification, expanded to eight grades, to better classify degenerated discs in elderly patients, in which most discs would be classified as Pfirrmann grade III or IV. These additional classifications are mainly based on the level of disc height loss. However, a severe limitation of these classification methods is the lack of correlation between observed IVD degeneration and LBP [9,10].

### 3.3. Modic Changes

In 1988, Modic et al. [35] classified the intensity changes on MRI of vertebral endplates and subchondral bone into three types. Modic type I involves low intensity on vertebral body endplates on T1-weighted imaging (T1WI) and high intensity on T2WI, which is suggested to reflect bone marrow edema and inflammatory changes. Type II is classified as highly intense vertebral endplates on both T1WI and T2WI, and it is thought to reflect fatty degeneration of bone marrow (Figure 2) [36]. Lastly, type III presents low intensity at both T1WI and T2WI and is reported to reflect sclerosis of the subchondral bone. In particular, Modic type I has been reported to be associated with LBP, intervertebral instability, and inflammation [37,38], although care should be taken as pyogenic discitis can present similar intensity changes [39]. In addition, upright weight-bearing MRI scans have shown a correlation between increased Modic I changes extension and increased pain in the standing position [40]. Although systematic reviews have shown an association between Modic changes and LBP, it should be noted that Modic changes are also present in cases without LBP [41].

### 3.4. High-Intensity Zone (HIZ)

The high-intensity zone (HIZ) is a hyperintense area in the posterior AF of lumbar IVD on MRI T2WI, which is thought to reflect secondary inflammatory changes caused by fissures in the AF and NP tissue protruding into the fissures [42]. We previously reported that, for a patient with severe LBP presenting with HIZ on MRI, temporary pain improvement was observed through discoblock in the same area. Moreover, LBP and imaging findings were improved by cauterization of the HIZ area under endoscopic view (Figure 3) [43]. To underline our findings, a meta-analysis by Fang et al. emphasized that the diagnostic value of the HIZ as a source of pain in discogenic LBP is relatively high [44]. On the other hand, HIZs have been reported to be present in 24–56% of asymptomatic patients [45]; as such, the relationship between HIZ and LBP remains controversial [46].

## 4. Quantitative MRI in IVD Degeneration

T2WI is currently the most commonly employed MRI modality to depict the abovementioned pathologies toward diagnosis of discogenic LBP. However, these MRI findings are often seen in patients without LBP, and it is difficult to make a definitive diagnosis of discogenic LBP using T2WI alone. Furthermore, the signal intensity obtained by conventional T2WI is merely a relative signal and, thus, does not measure the absolute values of intradiscal water content or ECM composition. Therefore, a more detailed and quantitative evaluation method is needed to determine the effect of IVD degeneration or regeneration in clinical studies and preclinical animal experiments. In recent years, sequences enabling more objective and reproducible assessment have emerged through the quantification and mapping of metabolite concentrations in IVDs. In addition, direct correlation with discogenic LBP has been reported in several sequences. Below, we provide an overview of upcoming quantitative MRI imaging methods.

### 4.1. T2 Mapping

T2 mapping is derived from quantification of the T2 relaxation times of multiple echo times, enabling the visualization of biochemical composition of the disc through color mapping (Figure 4). Changes in T2 are a result of water content and to what extent water is “bound” or “free”. When IVD degeneration occurs, water content decreases and collagen organization changes from a more randomly orientated mesh to more uniform structure with loss of microstructural elements and increased NP collagen polarity [47]. Therefore, the T2 values in the NP tend to decrease with age and degeneration [48,49]. Watanabe et al. [50] showed that axial T2 mapping may be more useful than the Pfirrmann classification in detecting early degeneration in healthy volunteers. It has also been reported that discs with herniations and annular tears have significantly lower T2 values than normal discs [51]. Ogon et al. [52] showed that, in L4/5 degenerated discs, T2 values of posterior AF were significantly lower in patients with chronic LBP than in asymptomatic controls, and there was a weak negative correlation between T2 values and a lumbar visual analog scale (VAS), as well as a weak positive correlation between T2 values and the Japanese Orthopedic Association Back Pain Evaluation Questionnaire (JOABPEQ) score in patients with chronic LBP. In addition, Bruno et al. [53] showed that T2 mapping may be a useful indicator to predict disc shrinkage and the clinical response to oxygen–ozone chemonucleolysis. On the other hand, it should be noted that decreased T2 values on T2 mapping are also present in asymptomatic patients. Furthermore, others have shown a strong correlation between T2 relaxation times and PG levels, type II collagen content, and histological degeneration scores in various animal models such as rabbits and goats [54,55,56], and these findings have been used in the development of regenerative medicine for IVD repair [57]. However, the caveat is that these models generally involve acute disc degeneration models and do not include assessments of discogenic pain. The impact of T2 mapping on pain has been poorly studied, and further research in large cohorts is needed.

### 4.2. Diffusion-Weighted MRI with ADC Mapping

The conventional T1 and T2WI methods focus on an atomic-scale phenomenon, so-called “proton motion”, whereas diffusion-weighted imaging (DWI) focuses on a relatively large-scale phenomenon, i.e., the diffusion of water molecules. In vivo, diffusion is generally limited in tissues with a large number of cellular components, whereas it more freely diffuses in regions with abundant intercellular substances. The difference in ease of diffusion is visualized by DWI (Figure 4). DWI is an essential examination for the diagnosis of acute cerebral infarction (reflecting cytotoxic edema) and is commonly used to detect malignant tumors (as high cellularity determines restricted diffusion), abscesses, pyogenic spondylitis, and other infection foci (as pus determines restricted diffusion) in orthopedics [58]. By acquiring the DWIs in two different magnetic strength fields, the apparent diffusion coefficient (ADC) can be calculated, and it is used for tissue-specific quantitative evaluation (Figure 5). Reports have shown that ADC values correlate negatively with the degree of IVD degeneration [59] and positively correlate with the glycosaminoglycan (GAG) and water contents in the NP [60]. Wu et al. [61] showed a more linear relationship between age increase with ADC decrease in their younger age group compared to T2, and ADC may be more sensitive in detection of age-related IVD changes than T2. We previously reported on the use of ADC for MRI assessment in clinical trials of human NP cell transplantation [62]. A recent study [63] investigating nutrient availability in the IVD using MRI ADC maps and a two-dimensional steady-state finite element model confirmed that the concentration of nutrients such as glucose and oxygen decreased with increasing distance from the CEP, while an increase in acidity was observed. Furthermore, the physiological microenvironment in the IVD became more deficient as degeneration progressed, suggesting that the solute concentration in the IVD correlated with disc height, disc area, and mean NP ADC values. So far, the relationship between DWI and discogenic LBP has not been clarified.

### 4.3. T1ρ Mapping

While T2 mapping focuses on proton motion in water molecules in general, T1ρ mapping is a method focusing on proton motion in water molecules whose motion is restricted by local high-molecular-weight molecules, such as PGs; as a result, it exhibits a good correlation with the PG content in the IVD and cartilage matrix [64,65]. Accompanying IVD degeneration, the PG content in the NP decreases and the proton momentum in the water molecules increases; therefore, the condition of the IVD can be quantitatively evaluated by quantifying this momentum (Figure 6) [66]. In articular cartilage degeneration, decreased PG concentration occurs first, after which the collagen alignment becomes irregular and generates morphological changes [67]. Therefore, T1ρ mapping may be superior to T2 mapping, which is affected by changes in the collagen sequence, in detecting early cartilage degeneration because it captures the decrease in PG concentration in the early stages of degeneration [68]. A prospective study by Borthakur et al. [69] comparing chronic LBP patients scheduled for discography with an age-matched cohort of asymptomatic volunteers found that T1ρ values were significantly lower in painful discs than in control and nonpainful discs. The receiver operating characteristic area under the curve for T1ρ was 0.91 in predicting painful discs, indicating that it could act as a novel minimally invasive biomarker acting as an alternative to discography for diagnosing painful discs. Similarly, Fenty et al. [70] demonstrated the potential of combining T1ρ MRI with disc height analysis to predict painful discs without provocative discography. Furthermore, it has been reported that, in patients with chronic discogenic LBP, T1ρ values decreased with the progression of degeneration and correlated with Pfirrmann classification, as well as clinical outcomes such as SF-36 and the Oswestry Disability Index (ODI) [71]. Although T1ρ mapping has been shown to capture early disc degeneration [72], it has also been reported to be useful in diagnosing discs associated with LBP [69,70,71]. These studies have several limitations, however, such as a small study cohort, cohort heterogeneity, inconsistent images, and lack of meta-analysis. Thus, further study is needed to confirm the promising outcomes currently reported. We expect that T1ρ modalities will become commonly applied evaluation tools in clinical practice and regenerative medicine development for discogenic LBP.

### 4.4. Sodium MRI

The sodium MRI method is focused on estimating the amount of PG by measuring the sodium concentration in the IVD. As the negative charge of the GAG molecules, particularly in the NP, attracts and retains cationic sodium, their detection offers a promising indicator for PG content in the IVD. In a bovine ex vivo study, Wang et al. [73] reported a strong correlation between measured sodium levels and PG levels. They suggested that sodium MRI may be useful as a noninvasive diagnostic tool for PG reduction occurring during the early stages of IVD degeneration. In asymptomatic volunteers, the normalized sodium signal intensity on sodium MRI was negatively correlated with the modified Pfirrmann score [74]. In a comparative rabbit study of T2 and sodium MRI [75], it was concluded that, although the sodium concentrations highly correlated with T2 values in degenerated discs, sodium changes were more sensitive to the differences in disc status compared to T2. Because T2 and sodium concentrations reflect different disc properties (T2: water and collagen content, sodium: PG content), performing both imaging approaches in the same session may be useful in understanding the process and mechanisms underlying IVD degeneration at the molecular level. However, the signal-to-noise ratio (SNR) of sodium MRI is significantly lower than that of conventional proton MRI [73], which means that either the spatial resolution must be sacrificed or the imaging time must be extended. The clinical application is also restricted due to the need for specialized imaging equipment. To date, no studies have reported on the usefulness of sodium MRI for identifying discogenic LBP.

### 4.5. Q-Space Imaging (QSI)

DWI is based on the theoretical assumption that the diffusion of water molecules shows a normal distribution, as would occur in an unobstructed space, and this diverges from the actual movement of water in vivo. In living organisms, several three-dimensional barriers exist, mainly due to structures such as cellular walls and ECM. Water molecules continuously collide with obstacles during movement, and, in most cases, a normal distribution does not apply. Accordingly, q-space imaging (QSI) is a method that can analyze the restricted diffusion within tissues by measuring the distance (displacement) that water molecules actually traversed by using additional data in comparison with ordinary DWI [76,77]. One of the limitations in the clinical application of QSI is the long imaging time, to enable spatial water mobility. Alternatively, diffusional kurtosis imaging (DKI) involves a sequence in which the degree of deviation from a normal distribution of diffusion is visualized using an index called “kurtosis”, and it has demonstrated advantages in the field of neurology due to its shorter imaging time compared with QSI [78,79]. In the orthopedic field, QSI has been reported for utilization in monitoring demyelination and remyelination after spinal cord injury and proved useful for visualization of spinal cord white matter [80]. Furthermore, its potential as a biomarker to capture diurnal variation in the IVD has been reported [81]. DKI has also been shown to be beneficial in detecting microstructural differences of Pfirrmann grade I and II discs in normal rats with relatively high accuracy [82]. Additionally, in a study of the inhibitory effect of the antioxidant *N*-acetylcysteine on a rat IVD degeneration model, QSI enabled the detection of slight changes that could not be detected by conventional quantitative methods, e.g., T2 mapping and ADC [83]. As such, QSI and DKI form promising techniques for evaluating IVD degeneration and regeneration, although practical considerations do require further development. Moreover, to date, no studies have examined the applicability of QSI outcomes with discogenic LBP.

### 4.6. Chemical Exchange Saturation Transfer (CEST)

Chemical exchange saturation transfer (CEST) imaging is a technique based on the chemical exchange of protons generated between hydroxyl and amide groups and bulk water contained in tissues, and it is able to detect, at high sensitivity, compounds at low concentrations that are difficult to detect directly [84,85]. gagCEST is a CEST method that can achieve contrasts that reflect GAG concentrations in vivo (Figure 6). In a study conducted on healthy volunteers using clinical 3T-MRI, gagCEST values in the NP correlated negatively with the Pfirrmann classification [66] and were reported to be significantly lower in herniated discs [86]. Furthermore, gagCEST values in several studies were significantly lower in patients with radiculopathy and LBP compared to patients without pain [87,88]. Whereas T2 mapping and T1ρ mapping based on changes in relaxation time are unable to directly evaluate GAG concentration, gagCEST can directly detect endogenous GAG concentrations in the IVD and does not require a sodium coil or special radiofrequency equipment; therefore, it is expected to face relatively easy clinical adaptability. Recently, quantitative CEST (qCEST) MRI was shown to have the potential to detect pH changes in IVD and to be capable of objectively and noninvasively diagnosing painful discs [89]. A clinical study of patients with chronic LBP also reported that the qCEST values of painful discs were significantly higher than those of nonpainful discs, and that the qCEST/T2 ratio was able to identify discs associated with pain with a sensitivity of 78% and a specificity of 81% [90]. Nonetheless, large-scale studies are needed to confirm these observations and are highly anticipated.

### 4.7. MR Spectroscopy (MRS)

The magnetic fields of atomic nuclei in molecules are affected by the surrounding environment, and a phenomenon called “chemical shift” based on the chemical bonds formed by the atoms causes a difference in resonance frequencies [91]. MRS can measure the status of individual metabolites in vivo by frequency separation of signals from various metabolites embedded in the water signal using signal intensity and chemical shift [92]. In a comparative study using ex vivo MRS of patients with discogenic LBP diagnosed by discography and patients with scoliosis, Keshari et al. [93] showed that decreases in the PG/lactate ratio and PG/collagen ratio are potential noninvasive spectroscopic biomarkers for identifying patients with discogenic LBP. Although a direct relationship between lactate concentration and pain has not yet been established, there are reports of an association between incision-induced pain behavior and increased lactate concentration at wound sites in rats [94], as well as data showing that neural activity of dorsal root ganglion neurons is enhanced by low pH [95]. Thus, low pH due to increased lactate may stimulate nerve fibers and correlate with discogenic pain. Gornet et al. [96] found that MRS scores, using structural integrity markers (carbohydrate/collagen and PG) expected to decrease with disc degeneration and acidic pain markers (alanine, lactate, and propionate) expected to increase with hypoxia and inflammation, could be obtained by quantifying spectral features from optimized clinical MRS data identified painful discs in patients with chronic LBP with a sensitivity of 82% and a specificity of 88% when compared to provocative discography. As MRS targets the measurement of trace metabolites with signal intensity much lower than that of MRI, the spatiotemporal resolution is low at current clinical MRI field strengths [97]. The widespread use of high-field MRI in the future may expand the clinical application of MRS, which can provide biochemical information.

### 4.8. Ultrahigh-Field MRI

Compared to the 1.5 T and 3 T MRI commonly used in clinical practice today, ultrahigh-field MRIs (7 T and 9.4 T) have been reported to have higher spatiotemporal resolution and to be able to provide more detailed morphological, biochemical, and functional information in musculoskeletal tissues [98,99]. In addition, 9.4 T MRI had significantly better inter- and intra-observer agreement in Pfirrmann classification compared to 3 T MRI, indicating that 3 T MRI tended to overestimate the degree of IVD degeneration [99]. In the US and Europe, clinical 7 T MRI was approved in 2017, but the high cost of introduction and maintenance still limits routine clinical use. Further studies are needed to demonstrate the diagnostic usefulness of ultrahigh-field MRIs.

## 5. Identification of Painful Discs

Although we have described various MRI sequences that can quantitatively evaluate IVD degeneration, the values obtained from these quantitative MRIs are not directly related to LBP. As mentioned above, although there are many reports that T2 mapping, T1ρ mapping, qCEST, and MRS are useful for identifying painful discs, these studies are limited in impact as they (1) have not been confirmed to represent actual histological changes, (2) involve heterogenous and small cohorts, (3) involve a wide range of imaging setups and settings making comparison between studies difficult, and (4) are ambiguous and inconsistent in the identifications and specifications of IVD diseases and its associated features. At present, only the invasive method of provocative discography or discoblock is available for the direct diagnosis of disc-derived pain. However, provocative discography has a high false-positive rate, especially in patients with psychological abnormalities [100], and it has been reported to accelerate disc degeneration [101]. The development and validation of these MRI sequences is, thus, highly anticipated, and they will hopefully provide a noninvasive diagnostic tool to identify painful discs, enable classification of in situ IVD regeneration, and further help classify different types of discogenic LBP.

## 6. Conclusions and Future Trends

In this paper, we described MRI techniques that enable quantitative analyses as a new alternative to the qualitative evaluation of IVD degeneration by commonly applied T2WI (Table 1). These quantitative MRIs have multiple practical limitations such as requiring the purchase of expensive programs and the limited number of equipment types capable of performing these sequences. Although some progress has been made in the sequences and techniques for qualitative and quantitative measurement of IVD degeneration and age-related changes in the substrate, there is a need for the dissemination of technologies with high specificity of clinical symptoms such as pain. In the future, further improvements in resolution and technologies are expected to enable the development of methods to detect subtle biochemical changes and inflammatory findings in IVD and to correlate these imaging observations with clinical symptoms. In addition, the relationship between brain function and chronic LBP offers a promising angle for future investigation [102], and the combination of functional MRI of the brain and quantitative MRI of the lumbar spine may further elucidate the pathology of chronic LBP. As future trends, artificial intelligence and deep learning are also expected to improve the diagnostic value of MRI [103]. Moreover, these quantitative MRIs have the potential to be applicable to the evaluation of the therapeutic effects of IVD regenerative medicine, the development of which is currently progressing in multiple aspects, and further development is expected in the future.

## Figures and Tables

**Figure 1 diagnostics-12-00707-f001:**
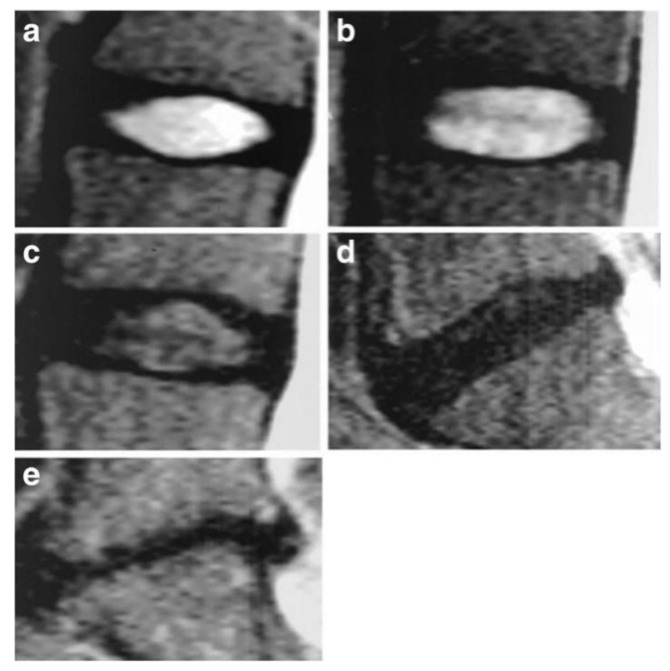
Pfirrmann classification (reused from [34]). A qualitative visual assessment method for lumbar disc degeneration using T2-weighted mid-sagittal images. Images represent the five Pfirrmann grades ranging from grade I (**a**) to V (**e**), representing healthy to severely degenerated IVD respectively. (**a**) Grade I: The structure of the disc is homogeneous with a bright white signal intensity and a normal disc height. (**b**) Grade II: The structure of the disc is not fully homogeneous with a clear boundary between nucleus pulposus (NP) and annulus fibrosus (AF), and a normal disc height. (**c**) Grade III: The structure of the disc is inhomogeneous with a moderately decreased white signal intensity, an unclear boundary between NP and AF, and an almost normal disc height. (**d**) Grade IV: NP signal intensity is darkened with no boundary between NP and AF, and a moderately decreased disc height. (**e**) Grade V: NP signal intensity is black with collapsed disc space.

**Figure 2 diagnostics-12-00707-f002:**
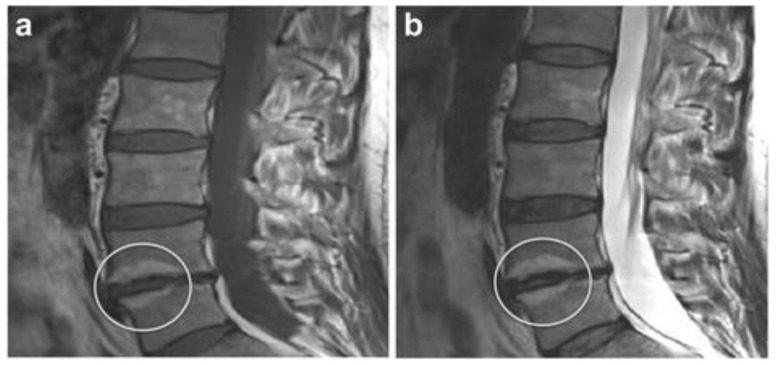
Modic type II change (reused from [36]). (**a**) T1-weighted image; (**b**) T2-weighted image. The white circle showing Modic type II change.

**Figure 3 diagnostics-12-00707-f003:**
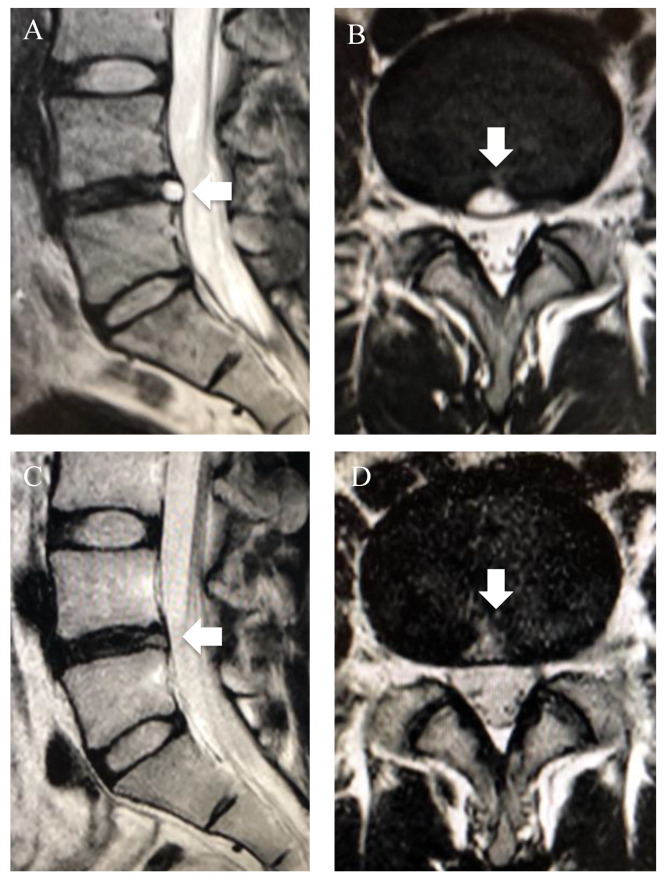
High-intensity zone (HIZ) on T2-weighted images (T2WI) (reused from [43]). Sagittal and axial T2WI of a 24 year old patient with a complaint of severe lower-back pain (LBP). (**A**,**B**) HIZ (white arrow) was observed in the midline of the posterior annulus fibrosus of the L4/5 disc. (**C**,**D**) Discoblock caused recurrent pain and temporary improvement of pain, and cauterization of the HIZ area under endoscopic view resolved LBP and showed improvement of the HIZ.

**Figure 4 diagnostics-12-00707-f004:**
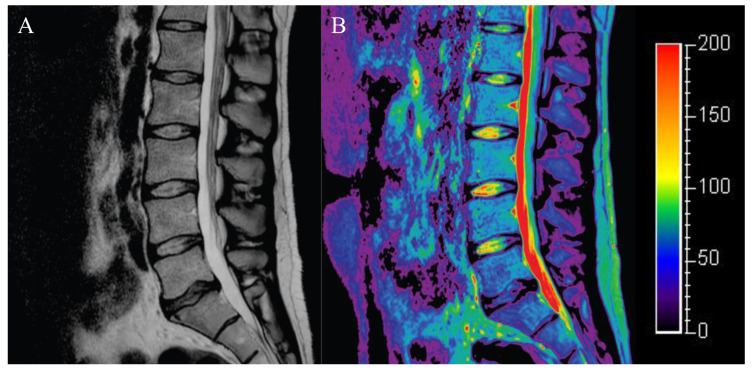
T2 mapping image. Lumbar spine MRI of an asymptomatic patient through (**A**) T2WI and (**B**) T2 mapping sequences. The T2 values of the L5/S disc, which is classified as Pfirrmann grade IV, also shows a decrease in T2 mapping intensity.

**Figure 5 diagnostics-12-00707-f005:**
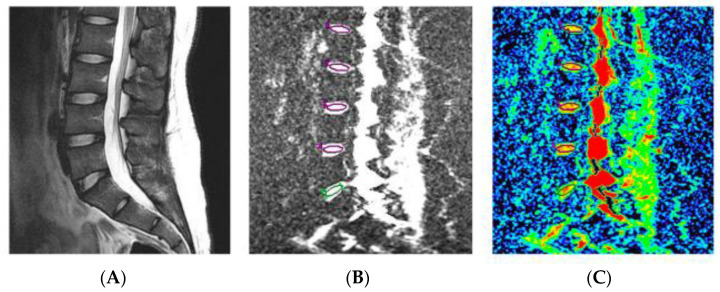
Diffusion-weighted image (DWI) and apparent diffusion coefficient (ADC) mapping image (reused from [61]). (**A**) T2WI; (**B**) DWI; (**C**) ADC mapping image.

**Figure 6 diagnostics-12-00707-f006:**
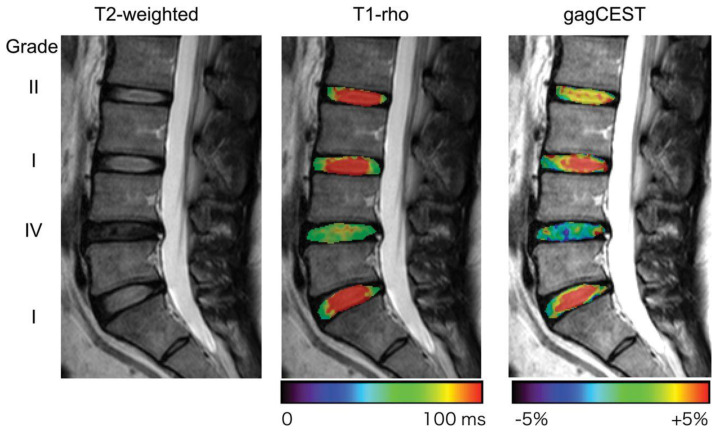
T1ρ mapping image and glycosaminoglycan chemical exchange saturation transfer (gagCEST) image (reused from [66]). The L4/5 disc, classified as Pfirrmann grade IV on T2WI, shows a shortened T1ρ and lower gagCEST signal intensity compared with the other discs without degeneration (Pfirrmann grade I or II).

**Table 1 diagnostics-12-00707-t001:** Summary of the characteristics of various MRI sequences as imaging diagnostic tools for IVD degeneration and painful discs.

Sequences	Measurement Target	Advantages	Disadvantages	Relation to Discogenic LBP
T2-weighted imaging	Water content and disc morphology	Most commonly used and well established	Not for quantitative evaluation Reproducibility	△ (Modic change and HIZ)
T2 mapping	Water content and to what extent water is “bound” or “free”	Allows for quantitative evaluation of water and collagen content	Not only sensitive to PG content	△
Diffusion-weighted MRI with ADC mapping	Diffusion of water molecules	Allows for evaluation of nutrient availability	Based on the theoretical assumption that the diffusion of water molecules shows a normal distribution	×
T1ρ mapping	PG content	Allows for detection of early stages of degeneration	Lack of standardization for imaging settings	△
Sodium MRI	PG content based on sodium concentration	High correlation with PG content	Low SNR Need for specialized hardware	△
Q-space imaging	Restricted diffusion of water molecules	Allows for detection of the degree of restricted diffusion of actual water molecules	Need for longer scan time Little available literature	×
gagCEST quantitative CEST	GAG content pH changes	Allows for detection of endogenous GAG content No need for specialized hardware	Low SNR Need for longer scan time	△
MR Spectroscopy	Chemical composition of the trace metabolites	Different characteristics from other MRI sequences	Low SNR Does not allow for identification of the source of each signal	△

Footnote: △ indicates that the relationship with discogenic LBP has been reported, while × indicates that it has not been reported. Abbreviations: MRI, magnetic resonance imaging; IVD, intervertebral disc; LBP, lower-back pain; HIZ, high-intensity zone; PG, proteoglycan; ADC, apparent diffusion coefficient; SNR, signal-to-noise ratio; GAG, glycosaminoglycan; CEST, chemical exchange saturation transfer.

## Data Availability

Data can be requested from the corresponding authors upon reasonable request.

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
