# Peer review of "Imaging Evaluation of Intervertebral Disc Degeneration and Painful Discs—Advances and Challenges in Quantitative MRI"

_diagnostics, 2022, doi:10.3390/diagnostics12030707_

Round 1

Reviewer 1 Report

It is a well written paper. I have some suggestions, see the file.

In introduction, insert about differential diagnosis of LBP and insert suggested articles

Reis F, Macedo E, França Junior MC, Amstalden EI, Appenzeller S. Retroperitoneal Ewing's sarcoma/embryonal tumor: a rare differential diagnosis of back pain. Radiol Bras. 2017;50(6):409-410.

Reis EP, Silva Junior NA, Appenzeller S, Reis F. Radicular compression syndrome after exercise in a young patient: not everything is a herniated disk!. Radiol Bras. 2018;51(6):408-409.

Structurally, the border between NP and AF blurs as the water content in the NP decreases with PG loss, leading to a low signal intensity on T2 WI and to a decrease in the disc height

A more detail description of Pfirrmann classification should be inserted (it is just illustrated in Figure 1)

on T1-weighted imaging (T1WI) and high intensity on T2WI,

When you explain about Modic degenerations and pyogenic discitis mention the role of paramagnetic contrast

DWI is an essential examination for the diagnosis of acute cerebral infarction (and reflects cytotoxic edema) and is commonly used to detect malignant tumors (the high cellularity determines restricted diffusion) abscesses, pyogenic spondylitis, and other infection foci (as pus determines restricted diffusion) in orthopaedics [56].

Consider expand the following topics;

Carbohydrate/collagen (CA) and PG regions as structural integrity markers expected to decrease with disc degeneration, and

Alanine (AL), LA, and propionate (PA) regions as acidic pain markers (e.g., from hypoxia, infammation, and/or Propionibacterium acnes infection) expected to increase with discogenic pain

Consider illustrate spectroscopy with a Figure and discuss about its limitations, how is the quantification?

Author Response

Author Response:

Dear Editors and Reviewers

Thank you very much for reviewing our manuscript and offering valuable advice.

We have addressed your comments with point-by-point responses, and revised the manuscript accordingly.

Title of the Manuscript: Imaging Evaluation of Intervertebral Disc Degeneration and Painful Discs - Advances and Challenges in Quantitative MRI

Manuscript ID: diagnostics-1618481

Reviewer: 1

Comment 1: In introduction, insert about differential diagnosis of LBP and insert suggested articles.

Reis F, Macedo E, França Junior MC, Amstalden EI, Appenzeller S. Retroperitoneal Ewing's sarcoma/embryonal tumor: a rare differential diagnosis of back pain. Radiol Bras. 2017;50(6):409-410.

Reis EP, Silva Junior NA, Appenzeller S, Reis F. Radicular compression syndrome after exercise in a young patient: not everything is a herniated disk!. Radiol Bras. 2018;51(6):408-409.

Authors’ Response: We appreciate the reviewer’s comment. However, the proposed articles are very rare cases and not directly related to discogenic low back pain. Thus, we consider it inappropriate for the purpose of our paper.

Comment 2: A more detail description of Pfirrmann classification should be inserted (it is just illustrated in Figure 1).

Authors’ Response: As you pointed out, we have added the following text to Figure 1 legend.

Change to Text: Grade I: The structure of the disc is homogeneous with a bright white signal intensity and a nor-mal disc height. Grade II: The structure of the disc is not fully homogeneous with a clear bounda-ry between nucleus pulposus (NP) and annulus fibrosus (AF), and a normal disc height. Grade III: The structure of the disc is inhomogeneous with a moderate decreased white signal intensity and an unclear boundary between NP and AF, and an almost normal disc height. Grade IV: NP signal intensity is darkened with no boundary between NP and AF, and a moderately decreased disc height. Grade V: NP signal intensity is black with collapsed disc space.

Comment 3: Consider illustrate spectroscopy with a Figure and discuss about its limitations, how is the quantification?

Authors’ Response: Although we tried to add a spectroscopy figure, the publisher refused to reuse the image. We described the limitations in the text and Table 1.

Reviewer 2 Report

Excellent review, but very brief. Would benefit from some added decription of various MRI technqiues, details of whats limiting clinical use, any future and whats the time frame if any. 

Reviewer: 2

Comment 1:  Excellent review, but very brief. Would benefit from some added decription of various MRI technqiues, details of whats limiting clinical use, any future and whats the time frame if any.

Authors’ Response: Thank you very much for your comment.

To the reviewer: Again, we wish to express our appreciation to the reviewers for your insightful comments on our paper. The comments have helped us significantly improve the paper. Please let us know if you require further clarification.

Respectfully yours,

Shota Tamagawa MD

Corresponding author’s name and complete contact information

Daisuke Sakai MD, PhD

Department of Orthopaedic Surgery, Surgical Science, Tokai University School of Medicine, 143 Shimokasuya, Isehara, Kanagawa 259-1193, Japan

Tel: +81-463-93-1121, Fax: +81-463-96-4404

E-mail: daisakai@is.icc.u-tokai.ac.jp